# Type 2 Diabetes Mellitus and Cancer: Epidemiology, Physiopathology and Prevention

**DOI:** 10.3390/biomedicines9101429

**Published:** 2021-10-09

**Authors:** Cristina Rey-Reñones, Jose Miguel Baena-Díez, Isabel Aguilar-Palacio, Cristina Miquel, María Grau

**Affiliations:** 1Research Support Unit-Camp de Tarragona, Catalan Institute of Health (ICS), 08908 Tarragona, Spain; crey.tgn.ics@gencat.cat; 2IDIAP Jordi Gol, Catalan Institute of Health (ICS), 08908 Barcelona, Spain; josemibaena@gmail.com; 3School of Medicine and Health Sciences, Universitat Rovira i Virgili, 43003 Reus, Spain; 4La Marina Primary Care Center, Catalan Institute of Health (ICS), 08908 Barcelona, Spain; 5Research Group in Health Services of Aragon, (GRISSA) IIS Aragón, University of Zaragoza, 50009 Zaragoza, Spain; iaguilar@unizar.es; 6Department of Medicine, University of Barcelona, 08036 Barcelona, Spain; cristina.miquel@gmail.com; 7Serra Húnter Fellow, Department of Medicine, University of Barcelona, 08036 Barcelona, Spain; 8Biomedical Research Consortium in Epidemiology and Public Health (CIBERESP), 08007 Barcelona, Spain; 9IMIM-Institut Hospital del Mar d’Investigacions Mèdiques, 08003 Barcelona, Spain

**Keywords:** type 2 diabetes mellitus, neoplasms, epidemiology, obesity, metformin, thiazolidinediones, disease prevention

## Abstract

Individuals with type 2 diabetes mellitus are at greater risk of developing cancer and of dying from it. Both diseases are age-related, contributing to the impact of population aging on the long-term sustainability of health care systems in European Union countries. The purpose of this narrative review was to describe, from epidemiological, pathophysiological and preventive perspectives, the links between type 2 diabetes mellitus and the most prevalent cancers in these patients. Multiple metabolic abnormalities that may occur in type 2 diabetes mellitus, particularly obesity, could explain the increased cancer risk. In addition, the effectiveness of drugs commonly used to treat type 2 diabetes mellitus (e.g., metformin and thiazolidinediones) has been broadly evaluated in cancer prevention. Thus, a better understanding of the links between type 2 diabetes mellitus and cancer will help to identify the contributing factors and the pathophysiological pathways and to design personalized preventive strategies. The final goal is to facilitate healthy aging and the prevention of cancer and other diseases related with type 2 diabetes mellitus, which are among the main sources of disability and death in the European Union and worldwide.

## 1. Introduction

How long we live and what proportion of that life is spent in good health have important implications for individuals and societies. Currently, one-fifth of the 512.4 million inhabitants of the 28 European Union countries (EU-28) are at least 65 years old—an increase of 0.3 percentage points in one year and 2.6 percentage points over the past decade. Moreover, recent projections suggest an increase in this population to 29.1% by 2080, reflecting an additional 53.3 million older inhabitants in the EU-28 [1].

The aging process is the main risk factor for the incidence and progression of type 2 diabetes mellitus and cancer, two of the most common causes of death and disability in the EU-28 and worldwide [2]. Researchers and policy makers agree that population aging is due to decreased fertility, particularly in developed countries, and increased life expectancy with no apparent plateau [3,4]. While modern medical innovations have increased longevity over the past century, the shift toward an older population at increased risk of diabetes and cancer [5,6] has the potential to generate a healthcare crisis in the coming years.

There are several biologically plausible explanations for the association between type 2 diabetes mellitus and cancer development. In cancers, malignant transformation typically involves a process of initiation followed by promotion and progression, stimulating cell growth and development. Patients with both diabetes and cancer share multiple common risk factors associated with initiation, including age, male or female gender depending on cancer type and location, obesity, limited physical activity and poor dietary and lifestyle habits [7]. In addition, the chronic effects of endogenous or exogenous hyperinsulinemia may promote malignant transformation via direct or indirect mechanisms [8]. At this point, scientific evidence shows that the adoption of a healthy lifestyle could substantially reduce premature mortality due to type 2 diabetes mellitus and its complications (e.g., cancers, cardiovascular diseases), as the beneficial effect of prevention is prolonged beyond the 8th decade of life [9,10]. However, the classic model of the determinants of health shows that individual lifestyles are embedded in social norms and networks and in living and working conditions, which in turn are related to the wider socioeconomic and cultural environment [11]. Therefore, the challenge of prevention of chronic diseases, such as type 2 diabetes mellitus and its associated complications, requires a recognition that success depends not only on the individual, but also on social and community networks and general socioeconomic, cultural and environmental conditions [12,13,14].

The objective of this narrative review was to consider the links between type 2 diabetes mellitus and cancer from three distinct perspectives: (1) Epidemiological, to describe the burden of cancer attributable to type 2 diabetes mellitus; (2) pathophysiological, to describe subjacent pathological mechanisms that increase the risk of cancer in individuals with type 2 diabetes mellitus; and (3) preventive, to describe potential preventive strategies aimed to prevent both type 2 diabetes mellitus and cancer.

## 2. Diabetes and Cancer: Burden of Disease

Diabetes constitutes a worldwide public health problem that affected 463 million people in 2019, considering all different subtypes of the disease; this number is projected to reach 578 million by 2030 and 700 million by 2045 [15]. The life expectancy of a 50-year-old individual with diabetes is 6 years shorter than it would be without the disease [16]. Type 2 diabetes mellitus is estimated to account for 90–95% of all cases [17], double the patient’s risk of cardiovascular events and increase 1.5-fold the risk of developing cancer compared with the general population [18]. In recent decades, reports of a negative annual percentage change in cardiovascular diseases, leading to a diversification of forms of diabetes-related mortality, have suggested implications for prevention, monitoring and clinical management of diabetes. A decreasing trend was also observed for cancers, but the magnitude of change was lower, compared to cardiovascular diseases. As a result, cancer surpassed cardiovascular diseases as the leading cause of death in individuals with type 2 diabetes mellitus in the Hong Kong Diabetes Surveillance Database in 2011 [19] (Table 1).

Several cohort studies have assessed the magnitude of the association and the risk of cancer in different locations. In 2011, the Emerging Risk Factors Collaboration published data on 123,205 deaths reported in 97 prospective studies (820,900 patients, 6% with diabetes) [17] and in Spain, the pooled analysis of the FRESCO Project (Spanish Risk Function of Coronary Events and Others) included 55,292 individuals from 12 population cohorts followed up for 10 years (15.6% with diabetes) [18]. In 2013, the Asia Cohort Consortium performed a pooled analysis of more than 20 cohorts representative of East and South Asian populations, including 771,297 individuals (4.7% with diabetes) [20]. Most recently, the National Health Research Institute in Hong Kong studied 895,434 individuals with diabetes and the same number of individuals without (1,790,868 participants) [21]. In all four studies, the results on the risk of death from cancer in different locations were highly consistent and can be summarized in two main concepts: (1) The risk of death from digestive cancers, particularly involving the liver, colon-rectum and pancreas, was higher in individuals with diabetes compared to those without diabetes; and (2) in women, diabetes was significantly associated with the risk of death from breast and ovarian cancers (Figure 1). Nevertheless, it remains unclear whether type 2 diabetes mellitus is causally related to cancer or the observed association is confounded by other factors.

**Table 1 biomedicines-09-01429-t001:** Trends in cause-specific death in individuals with diabetes mellitus.

Study	Country	Time Period	Gender	Age	Annual Percent Change	
					CVD	Cancer
National Health Interview Survey [22]	USA	1985–2015	All	≥20	–3.32	–1.57
National Diabetes Service [23]	Australia	2000–2011	Female	0–40	−6.65	−7.15
				40–60	−5.10	−1.55
				60–85	−6.35	−4.21
			Male	0–40	−1.24	−2.39
				40–60	−3.94	−1.40
				60–85	−5.43	−4.42
Hong Kong Diabetes Surveillance [19]	Hong-Kong	2001–2016	Female	45–59	−6.20	−5.60
				60–74	−12.50	−7.70
				≥75	−9.00	−5.30
			Male	45–59	−6.40	−7.40
				60–74	−9.90	−6.90
				≥75	−9.20	−6.00

CVD: Cardiovascular disease.

## 3. Pathophysiological Mechanisms That Explains the Link between Cancer and Diabetes

Metabolic abnormalities observed during the onset and progression of diabetes (Figure 1) may have a critical role in the initiation and progression of carcinogenesis [24]. The supraphysiological concentrations of both insulin and glycemia to which body tissues are exposed constitute a potent growth factor and energy source, respectively, that are essential for neoplastic transformation and cancer progression.

### 3.1. Hyperglycemia

Hyperglycemia is responsible for induction of oxidative stress and DNA damage, which may trigger the first phases of tumorigenesis [25,26]. In vitro studies conducted on tumor cell lines have shown that high glucose concentrations are able to modify the expression of genes related to proliferation, migration and cell adhesion [27]. Hyperglycemia may also contribute to the generation of advanced glycation end products (AGEs) that stimulate production of reactive oxygen species and inflammation. Chronic activation of the AGEs pathway has been shown to promote tumor transformation of epithelial cells and resistance of tumor cells to oxidative stress [28,29]. Moreover, during cancer progression, tumor cells are known to switch their energy source from mitochondrial oxidative phosphorylation to a less efficient glucose-dependent glycolytic pathway. This phenomenon, still not completely clarified, is known as the ‘‘Warburg effect” and has been recognized as a hallmark of virtually all types of cancer. Consequently, tumor cells require increased glucose uptake to provide energy and continue their proliferation [30,31].

### 3.2. Insulin

Endogenous hyperinsulinemia, mainly observed in type 2 diabetes mellitus patients in response to reduced insulin sensitivity of peripheral tissues, may also contribute to the relationship between diabetes and cancer by activating insulin receptors, IGF-1 receptors and hybrid insulin/IGF-1 receptors, all of which can stimulate cancer cell proliferation and survival and promote metastasis, thus favoring cancer progression [24]. Insulin stimulation from increased expression of insulin receptors may result in enhanced proliferation with loss of cell contact inhibition. Indeed, cancer cells frequently show augmented insulin receptor expression levels, mostly of the isoform A (lacking exon-11), whose activation is more responsible for mitogenic than metabolic effects and also shows high affinity for IGF-2 [32]. Moreover, high endogenous insulin concentrations observed in patients with type 2 diabetes mellitus are associated with reduced hepatic synthesis of IGFBP-1 (the main IGF-1 binding globulin), leading to increased bio-availability of IGF-1 and possible enhancement of its known mitogenic effects [8,24].

### 3.3. Inflammation

Chronic inflammation may establish a tumor-supporting micro-environment favoring the neoplastic process [33], characterized by high levels of oxidative stress and reactive oxygen species, abnormal adipokine production and activation of pro-inflammatory pathways. Increased circulating levels of specific adipose tissue-derived cytokines have been reported to promote tumor cell growth (tumor necrosis factor-a [TNF-a], interleukin [IL]-6), enhance metastasis (TNF-a, IL-6, transforming growth factor [TGF]-b), increase angiogenesis (TNF-a, IL-17, TGF-b) and impair the function of macrophages and NK cells [24,33].

Oxidative stress is another player tightly linked with inflammation and cancer risk in diabetes. Patients with diabetes usually have increased levels of all biomarkers of oxidative stress, which can mediate damage to cell structures, including lipids, membranes, proteins and especially DNA [34]. DNA mutation is a critical step in carcinogenesis, and elevated levels of oxidative DNA lesion markers, such as increased 8-hydroxy-guanosine (8-OH-G), have been noted in various tumors, strongly implicating this type of damage in the etiology of various cancers [35].

## 4. Obesity: A Common Risk Factor for Cancer and Diabetes

The scientific evidence points out excess weight as a common risk factor for both type 2 diabetes mellitus and cancer [36]. Obesity is a worldwide public health problem due to its high prevalence, the epidemic increase observed in recent decades and the associated diseases [37]. The World Health Organization (WHO) reports that more than 1.9 billion adults were overweight in 2016, of which 650 million were obese, doubling the worldwide prevalence of obesity since 1980 [38]. Recent projections are that global obesity prevalence will reach 18% in men and surpass 21% in women by 2025; severe obesity will exceed 6% in men and 9% in women [39]. Finally, overweight and obesity, as estimated with body mass index, have been associated with decreased functional ability and health status and increased risk of chronic conditions in classical observational studies [40]. Thus, obesity and type 2 diabetes mellitus are frequently associated with metabolic abnormalities that may contribute to cancer progression [8]. For instance, in the Asia-Pacific Cohort Studies Collaboration, the cancer locations with increased mortality risk in participants with obesity compared to those with normal weight were colon-rectum, breast (in women aged 60 years or older), ovary, cervix, prostate and blood (leukemia) [41]. Similar results were observed in a previous meta-analysis that analyzed data from 221 studies [42]. Considering that an estimated 90% of type 2 diabetes mellitus is attributable to excess weight [43], the most common cancer sites described in individuals with obesity are, not surprisingly, the same as in individuals with diabetes [17,18,20,21].

Multiple factors potentially contribute to the progression of cancer in obesity and type 2 diabetes mellitus, including hyperinsulinemia and insulin-like growth factor I, hyperglycemia, dyslipidemia, adipokines and cytokines and the gut microbiome. These metabolic changes may contribute directly or indirectly to cancer progression [8]. Thus, obesity, and specifically abdominal obesity, is associated with increases in adipose tissue inflammation from the production of cytokines and in the circulating concentrations of adipokines (produced by fat). These changes can contribute to insulin resistance in metabolic tissues (fat, liver and skeletal muscle), requiring increased production of insulin from the pancreatic beta cells to maintain normal glucose levels. This in turn leads to circulating hyperinsulinemia, which increases liver-derived insulin-like growth factor I (IGF-I) and alters circulating concentrations of IGF-binding proteins (IGF-BPs). Changes in IGFBPs in response to insulin may also occur in individual tissues, affecting the local concentration of bioavailable IGF-I [8,36,44,45,46]. This reaction could reduce apoptosis and increase cell proliferation in target cells, leading to tumor development [47,48,49]. Insulin itself is also known to have mitogenic properties [50]. Although the results of the FRESCO cohort did not show higher mortality from pancreatic cancer in individuals with diabetes [18], the liver and pancreas are among the tumor locations with the highest risk of cancer death in individuals with diabetes (Figure 2) in the other studies analyzed [17,20,21]. These two organs are likely to be exposed to a high level of endogenously produced insulin [36].

Insulin resistance is also associated with lipid abnormalities, characterized by elevated very low-density lipoproteins (VLDL) and triglycerides, decreased high-density lipoprotein (HDL) cholesterol and a decline in the hepatic production of sex hormone binding globulin (SHBG), which may lead to an increase in the levels of free hormones (including estrogen and testosterone). Furthermore, excess adiposity may lead to increased local aromatization of androgens to estrogens, which may affect tumor growth [8,36]. This pathway may also explain higher mortality risk from breast and ovarian cancers in women with diabetes compared to those without [17,18,20,21,51]. Indeed, a previous study underlined sex-related differences in the role of above-normal BMI, with obesity being particularly malignant in women. Thus, the hazard ratio of cancer death was 3.98 (95% confidence interval 1.53–10.37) for those with overweight and 11.61 (1.93–69.72) in those with obesity compared to those with normal weight—even higher than for cardiovascular disease (2.34 (1.19–4.61) and 5.65 (1.54–20.73) for overweight and obesity, respectively) [52].

Several epidemiological studies have explored the relationship between the clinical stages of hyperinsulinemia and the risk of cancer before diabetes onset [53,54,55]. Ontilio et al. found a 16% increase in the risk of breast cancer 10 years before the diagnosis of diabetes in the US population [54]. The same author described 28% higher risk of colon cancer in men in the pre-diabetes phase [55], in comparison with those who did not develop diabetes. On the other hand, a study conducted in Canadian population assessed the temporal relationship between diabetes and the risk of cancer in individuals with and without diabetes. This risk was evaluated at three time points: 10 years before diabetes was diagnosed and 3 months and 10 years after diagnosis [56]. Patients with diabetes were more likely (23%) to have cancer during the 10 years before diagnosis and the most common locations were the liver and the pancreas. An increased risk was also observed in the first 3 months after diagnosis, with no increased trend in the following 10 years. Other studies have corroborated these findings [57,58]. The increased risk of cancer after diabetes onset supports the potential role of hyperinsulinemia in this association, particularly in the prediabetes stage. Indeed, many years can elapse in the latent phase of cancer between the existence of diabetes risk factors and the development of overt cancer disease.

Finally, a review studied the relationship between the generation of lactic acid and the subsequent acidification of the surrounding environment. The findings could explain the negative prognosis of cancer in cases of extracellular lactic acidity. In diabetes, acidity affects insulin capacity to join the receptor, increasing the peripheric resistance and exacerbating symptoms. As a result, cancer progression acquires high invasive capacity and metastasis together with an inhibition of immune surveillance. Thus, acidity could be a relevant therapeutic objective [59].

## 5. Insulin Sensitizers and Cancer

With the introduction of metformin in the mid-1990s and the subsequent advent of thiazolidinediones, the treatment paradigm of type 2 diabetes mellitus changed. These drugs provided an opportunity to address and directly reverse, at least in part, the defects in insulin action seen in individuals with type 2 diabetes mellitus [60]. Biguanide metformin and the thiazolidinediones are effective antihyperglycemic agents with different modes of action. Metformin primarily reduces hepatic glucose, whereas thiazolidinediones primarily increases insulin sensitivity [61]. The effects of such drugs in cancer prevention have been broadly evaluated, together with those of sulfonylureas and statins.

### 5.1. Metformin

Metformin is the first-line type 2 diabetes mellitus treatment that works by reducing insulin resistance and fasting plasma insulin levels, leading to a reduction in blood glucose concentrations without causing overt hypoglycemia [12]. Several observational studies have shown that metformin may also reduce cancer risk and improve the prognosis and survival of cancer patients [62,63,64]. The plausible mechanisms that explain this association focus largely on inhibiting growth stimuli and metabolic processes within cancer cells and can be divided into insulin-dependent and -independent mechanisms that alter cancer cell growth [65].

The insulin-dependent mechanism is based on metformin’s effect on lowering serum levels of insulin and IGF-1, thus reducing the stimulus for growth. Given the potential importance of the Warburg effect in cancer metabolism, a glucose-rich environment could provide favorable conditions for aerobic glycolysis [30]. Treatment with metformin would therefore reduce hyperglycemia and any associated growth advantage in susceptible tumors [65].

Plausible insulin-independent mechanisms also exist. First, activation of LKB1/AMPK signaling by metformin could inhibit aerobic glycolysis in cells containing functional LKB1/AMPK pathways. In cells lacking functional LKB1/AMPK pathways, metformin could also induce tumor cell death by reducing ATP levels, which makes susceptible cells unable to respond to energy stress [66]. Thus, the anticancer effects of metformin have also been linked to decreasing hepatic gluconeogenesis by activating or phosphorylating AMP kinase (AMPK) in the liver [67,68]. Second, metformin may influence chronic inflammation, which can be an important factor in the initiation and promotion of carcinogenesis. Obese subjects characteristically develop a chronic proinflammatory environment with increased infiltration of immune cytokines such as leptin, adiponectin, interleukin 1 beta (IL-1β), IL-6, plasminogen activator inhibitor-1 (PAI-1) and tumor necrosis factor alpha (TNFα), all of which are associated with cancer proliferation and progression [69]. Thus, AMPK activation appears to inhibit the synthesis of proinflammatory cytokines in a variety of cell types, including macrophages and adipocytes. This suggests that metformin could potentially target proinflammatory cytokines within the tumor microenvironment, inhibiting growth in susceptible cancers [70]. Finally, metformin may have important effects in limiting tumor growth and metastasis by inhibiting endothelial cell migration and angiogenesis via AMPK-dependent reductions in growth factors, including vascular endothelial growth factor and PAI-1 [49,65,71].

A review of the most recent meta-analyses showed divergent results based on cancer locations. On the one hand, metformin treatment in patients with diabetes significantly reduced the risk of colorectal [72,73,74,75], hepatic [76,77], head and neck [78] and lung cancers [79,80]. Interestingly, recent experimental studies have shown that metformin may heighten the effectiveness of molecular-targeted agents for the treatment of non-small-cell lung cancer [81,82,83]. On the other hand, the effect of metformin on the risk of gynecological cancer was controversial. One meta-analysis found that this drug had potential anticarcinogenic effect [84], while Chu et al. did not find significant results in endometrial cancer [85]. Finally, metformin showed no effect on prostate [86,87,88] and bladder cancers [89] (Table 2).

New studies have shown the potential role of metformin in pathologies having common physiopathological pathways with cancer. For instance, AMPK pharmacological activation by metformin plays a key role in inhibiting inflammation and improving endothelial dysfunction [90]. Additionally, the anti-aging potential of metformin remains a highly debated topic, as it is unclear whether the ability of the drug to target multiple pathways of aging involves direct effects on multiple aging regulators or reflects downstream consequences of a primary action on a single mechanism [91].

### 5.2. Thiazolidinediones

Additionally, known as glitazones, thiazolidinediones are the most important intracellular agonists of peroxisome proliferator-activated receptors γ (PPARγ) and are used to reduce hyperglycemia in patients with type 2 diabetes [92]. PPARγ is a nuclear receptor and forms a heterodimer with retinoid X receptor, regulating the expression of genes involved in insulin action, adipocyte differentiation, inflammation and lipid metabolism. The activation of PPARγ due to the effect of thiazolidinediones leads to the differentiation of adipocytes, decreased release of free fatty acids from adipocytes and decreased production of prostaglandins, TNFα, IL-6, leptin and resistin by adipocytes. In addition, adiponectin production and glucose disposal by adipocytes increases, producing a rise in insulin sensitivity [93].

Shafiei-Irannejad et al. reviewed several studies showing that thiazolidinediones can be used alone as adjuvant therapy or in combination with other chemotherapeutics to sensitize tumor cells to the cytotoxic effects of anticancer agents [92]. Nevertheless, the results of the most recent meta-analyses show divergent results, particularly related to cancer locations. Thus, treatment with pioglitazone, one of the drugs from this group, was associated with mild risk of bladder cancer in most studies [94,95,96,97], whereas the risk of colorectal cancer decreased in those treated with thiazolidinediones [98]. No association was found between treatment with thiazolidinedione and the risk of breast cancer [99] (Table 3). Finally, Ciaramella et al., have highlighted the antitumoral effect of thiazolidinediones in individuals with non-small-cell lung cancer. This effect, explained by the inhibition of cancer cells’ growth and invasion through the modulation of bioenergetics and of cancer metabolism in both in vitro and ex vivo models, should be explored in future clinical studies [100].

## 6. Association between Cancer Risk and Other Drugs Used in Type 2 Diabetes Mellitus

### 6.1. Sulfonylureas

There are no conclusive results about the risk of cancer associated with sulfonylureas, a group of drugs that act by increasing insulin release from the beta cells in the pancreas [101]. Although no direct tumorigenic properties have been described, two studies have pointed out high risk of cancer in individuals treated with sulfonylureas compared to those treated with glyburide [102] or insulin [103]. In contrast, other studies found no effect [104] or a protective effect [105]. This uncertainty highlights the search for appropriate comparison groups as the current challenge in evaluating the effectiveness of sulfonylureas.

### 6.2. Statins

The antitumoral capacity of statins has been highlighted; however, the results are not conclusive [106]. A Japanese cohort study showed that patients with type 2 diabetes mellitus who were treated with statins had lower cancer incidence and mortality in the multivariable analysis, 10.5 and 3.7 per 1000 person-years, respectively, compared to 16.8 and 6.3 person-years, respectively, in the group with no treatment (Hazard ratio for cancer incidence = 0.67; 95% confidence interval 0.49–0.90, *p* = 0.007 and Hazard ratio for cancer mortality = 0.60; 95% confidence interval 0.36–0.98, *p* = 0.04) [107].

## 7. Prevention with Healthy Lifestyles

The social and economic consequences of type 2 diabetes mellitus are felt by all countries in terms of decreasing productivity and increasing poverty [15]. Prevention carries the greatest potential to reduce the burden of type 2 diabetes mellitus and the associated comorbidities [13,18]. The goal at the population level is to lower the mean glycemic level and reduce other cardiometabolic risk factors (e.g., blood pressure, total cholesterol) and shift the distribution of exposure in a favorable direction [108]. Monitoring of cardiometabolic risk factors seems particularly appealing to improve lifestyle (i.e., physical activity, healthy nutrition and smoking cessation) and thus improve not only cardiometabolic health, but also the risk of chronic diseases such as cancer [13,109]. For instance, the burden of potentially modifiable risk factors has been positively correlated with the individual’s perceived need to improve his or her physical health [110].

The increasing adoption of personalized eHealth solutions (i.e., health services and information delivered or enhanced through Internet of Things technologies) encourages individual responsibility for health-related decisions and for appropriate self-care and self-management of health conditions [111]. Some initiatives that applied tailored recommendations have shown important achievements in the control of cardiometabolic risk factors together with the development of healthy lifestyles [112,113]. These results suggest that interventions with tailored dietary recommendations that tap into individual beliefs about health problems and self-management could contribute to a successful lifestyle change and lead to improvements in the individual’s quality of life.

## 8. Summary

Individuals with type 2 diabetes mellitus are at greater risk of developing cancer and of dying from it. Multiple metabolic abnormalities that can occur in type 2 diabetes mellitus may explain the increased risk. Understanding more about the links between these two diseases, particularly the role of obesity, could help to identify the contributing factors. In addition, this knowledge will help to design personalized preventive strategies. The final goal is to facilitate healthy aging and the prevention of cancer and other diseases related with type 2 diabetes mellitus, which are among the main sources of disability and death in the EU and worldwide.

## Figures and Tables

**Figure 1 biomedicines-09-01429-f001:**
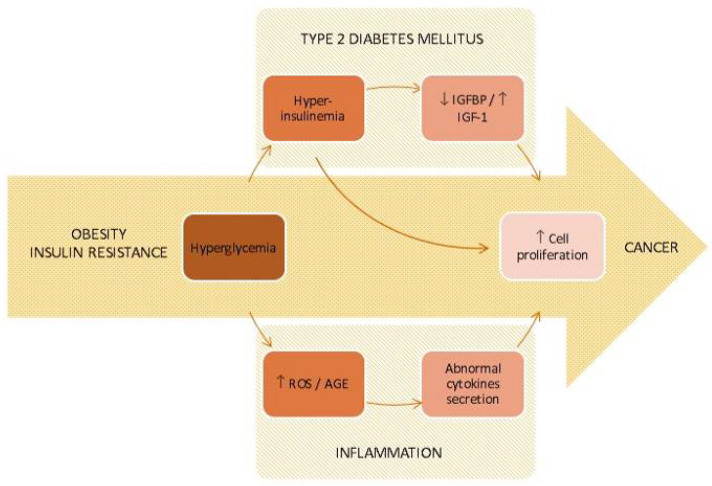
Pathophysiological links between obesity/insulin resistance, type 2 diabetes mellitus, inflammation and cancer. Adapted from Cignarelli et al. [24].

**Figure 2 biomedicines-09-01429-f002:**
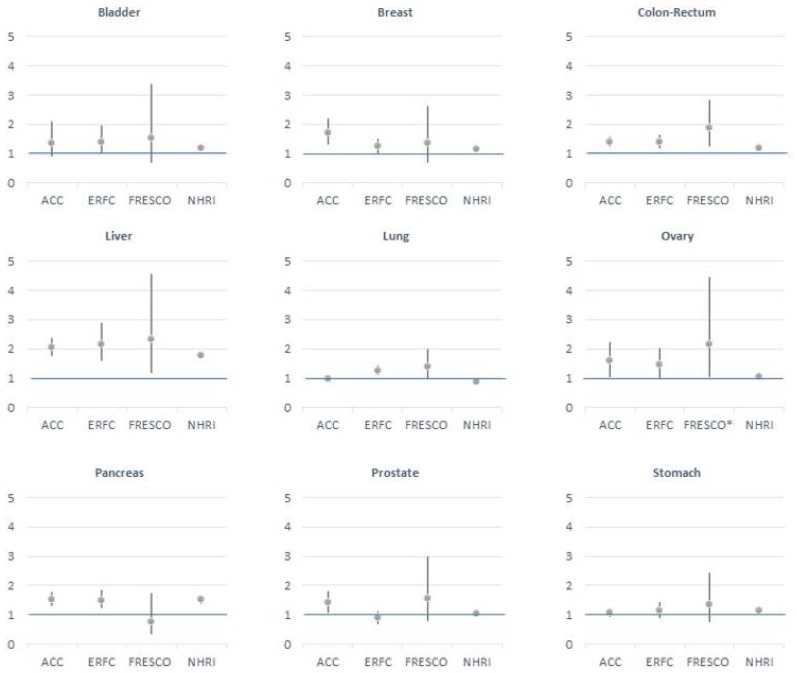
Hazard ratios for death from cancer in high-risk locations in individuals with type 2 diabetes mellitus. Data extracted from the Asia Cohort Consortium (ACC) [20], Emerging Risk Factors Collaboration (ERFC) [17], Spanish Risk Function of Coronary Events and Others (FRESCO) [18] and National Health Research Institute (NHRI) [21]. * Includes all cancers located in female genital organs.

**Table 2 biomedicines-09-01429-t002:** Meta-analyses of the effect of metformin on the risk of cancer in the last 5 years.

First Author and Publication Date	Cancer Location	Results	Conclusion
Yang, 2020 [72]	Colorectal	Metformin treatment significantly reduce the incidence of colorectal cancer in diabetic patients (adjusted RR = 0.884, 95% CI = 0.829–0.943).	Metformin therapy was associated with a significantly reduced risk of colorectal disease in patients with diabetes.
Jung, 2017 [73]	Colorectal	Metformin use reduced the risk of colorectal adenoma (pooled OR = 0.76, 95% CI = 0.63–0.92).	Metformin use seemed to be associated with a reduced risk of colorectal adenoma.
Liu, 2017 [74]	Colorectal	Metformin therapy was found to be associated with a decreased incidence of colorectal adenomas (adjusted OR = 0.75, 95% CI: 0.59–0.97).	Metformin therapy may be associated with a decreased risk of colorectal adenomas and colorectal cancer in type 2 diabetes mellitus patients.
Hou, 2017 [75]	Colorectal	Metformin therapy decreased the risk of colorectal adenoma (OR = 0.73; 95% CI = 0.58–0.90).	Metformin therapy was correlated with a significant decrease in the risk of colorectal adenoma in type 2 diabetes mellitus patients.
Ma, 2017 [76]	Liver	Metformin use reduced the risk of liver cancer (OR = 0.52; 95% CI = 0.40–0.68) compared with nonusers.	A protective effect for liver cancer was found in diabetic metformin users.
Zhou, 2016 [77]	Liver	The use of metformin was associated with a significant reduction of hepatocellular cancer risk (Risk Ratio = 0.49, 95% CI = 0.25–0.97).	Metformin was an effective strategy to reduce hepatocellular cancer risk.
Xiao, 2020 [79]	Lung	Metformin treatment was associated with decreased lung cancer incidence (HR = 0.78; 95% CI = 0.70–0.86).	Metformin was significantly associated with a decreased risk of lung cancer.
Yao, 2019 [80]	Lung	Compared to non-metformin users, metformin decreased lung cancer incidence in diabetic patients (RR = 0.89; 95% CI = 0.83–0.96).	Metformin use was related to a lower lung cancer risk in diabetic patients compared to nonusers.
Saka Herran, 2018 [78]	Head and neck	Metformin exerts significant beneficial effects on head and neck cancer risk (RR = 0.71, 95% CI = 0.61–0.84).	Metformin appeared to have beneficial effects on the risk of head and neck cancer
Wen, 2019 [84]	Gynecological cancers	Metformin may reduce the risk of gynecological cancers (RR = 0.49, 95% CI = 0.29–0.82).	Metformin can be used as a potential anticarcinogenic drug to prevent gynecological cancer.
Chu, 2018 [85]	Endometrial	Metformin was not significantly associated with a lower risk of endometrial cancer (OR = 1.05, 95% CI = 0.82–1.35).	Metformin was not beneficial for preventing endometrial cancer.
Wang, 2020 [86]	Prostate	No significant association between metformin and the risk of prostate cancer was found: - Case–control studies (pooled OR = 0.97, 95% CI = 0.84–1.12). - Cohort studies (pooled HR = 0.94, 95% CI = 0.79–1.12).	Metformin therapy was not associated with the risk of prostate cancer in patients with type 2 diabetes mellitus.
Feng, 2019 [87]	Prostate	Metformin use was not significantly associated with the risk of prostate cancer (RR = 0.97, 95% CI = 0.80–1.16).	No association was found between metformin use and prostate cancer risk.
Chen, 2018 [88]	Prostate	There was no association between metformin and prostate cancer (RR = 1.01, 95% CI = 0.86–1.18).	No association between metformin and risk of prostate cancer was identified.
Hu, 2018 [89]	Bladder	Metformin intake was not associated with a decreased incidence of bladder cancer (HR = 0.82, 95% CI = 0.61–1.09).	Metformin did not decrease the risk of bladder cancer.

CI: Confidence Interval. HR: Hazard Ratio. OR: Odds Ratio. RR: Relative Risk.

**Table 3 biomedicines-09-01429-t003:** Meta-analyses of the effect of thiazolidinediones on the risk of cancer in the last 5 years.

First Author and Publication Date	Cancer Location	Results	Conclusion
Liu, 2018 [98]	Colorectal	Decreased risk of colorectal cancer in patients treated with thiazolidinediones: (RR = 0.91; 95% CI = 0.84–0.99).	Protective association between thiazolidinediones use and the risk of colorectal cancer in patients with diabetes mellitus.
Du, 2018 [99]	Breast	No significant associations of thiazolidinediones use and risk of breast cancer: - Randomized controlled trials (pooled risk ratio = 0.77, 95% CI = 0.39–1.53). - Case–control studies (pooled OR = 0.99, 95% CI = 0.76–1.28). - Cohort studies (pooled RR = 0.94, 95% CI = 0.86–1.03).	No significant association was found between thiazolidinediones use and risk of breast cancer among diabetic women.
Tang, 2018 [94]	Bladder	Increased risk of bladder cancer in patients treated with pioglitazone: - Randomized controlled trials (OR = 1.84; 95% CI = 0.99–3.42). - Observational studies (OR = 1.13; 95% CI = 1.03–1.25).	Pioglitazone may increase the risk of bladder cancer, possibly in a dose- and time-dependent manner.
Yan, 2018 [95]	Bladder	Divergent results in the association between pioglitazone and bladder cancer according to the study design: - Randomized controlled trials showed no significant association. - Observational studies showed a significant association (pooled RR= 1.14, 95% CI = 1.03–1.26).	Pioglitazone is associated with an increased risk of bladder cancer.
Davidson, 2018 [96]	Bladder	No significant association between ever vs. never use of pioglitazone (HR = 1.09; 95% CI = 0.98–1.21).	The potentially very small number of patients at risk for bladder cancer after longer exposure to pioglitazone is far out-weighed by the much larger number of patients with cardiovascular disease and non-alcoholic steatohepatitis who would benefit from the drug.
Li, 2017 [97]	Bladder	Increased risk of bladder cancer in patients treated with pioglitazone (HR = 1.16, 95% CI = 1.06–1.25).	Pioglitazone use among subjects with diabetes mellitus increases mildly the risk of bladder cancer.

CI: Confidence Interval. HR: Hazard Ratio. OR: Odds Ratio. RR: Relative Risk.

## Data Availability

Data sharing not applicable.

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
