# Peer review of "Type 2 Diabetes Mellitus and Cancer: Epidemiology, Physiopathology and Prevention"

_biomedicines, 2021, doi:10.3390/biomedicines9101429_

Round 1

Reviewer 1 Report

The review deals with a very hot topic. Although many other similar reviews are present in the literature, it can be interesting and useful for the reader. The references are up to date and the tables and figures are clear.

However, this review raises some issues that the authors have to address.

1- In abstract, the authors state that the purpose of the review is the description of the relationship between diabetes and cancer "from the epidemiological, pathophysiological and preventive perspectives". Actually, the space dedicated to pathophysiological mechanisms in the manuscript is small and is limited to chapter 2 “Obesity: a common risk factor for cancer and diabetes”, which mainly deals with the role of insulin resistance. According to the title of the review too, the authors should insert in the text a paragraph in which the pathophysiological mechanisms that explain the high prevalence and incidence of cancer in diabetic subjects are more fully described.

2- Furthermore, it would be useful for the reader to have a figure outlining the main pathophysiological mechanisms linking cancer and diabetes.

3- In recent years, there has been growing interest in the possible role of metformin and also of pioglitation in non-small cell lung cancer. These molecules had not been studied in this type of cancer, which therefore could benefit from some drugs traditionally used in diabetes, as seen in both some experimental studies and initial clinical trials. In detail:

a- A major study described how metformin increases the antitumor activity of MEK inhibitors through downregulation of GLI1 in LKB1 positive human NSCLC tumor cells (Oncotarget. 2016 Jan 26; 7 (4): 4265-78. doi: 10.18632 / oncotarget.6559.), as well as an interesting review on the rationale for the use of metformin in lung cancer therapy (Expert Opin Investig Drugs. 2013, 22 (11): 1401-1409. doi: 1517/13543784.2013.828691).

b- In particular, in view of the lack of RCTs in this field, it would be very useful to cite the design of the METAL study (ESMO OPEN, 2017, 2 (2): e000132. Doi: 10.1136 / esmoopen-2016-000132).

c- Very recently, other insulin-sensitizer drug, PPAR-γ agonist pioglitazone, seems to positively act on NSCLC by suppression of cell growth and invasion via blockade of MAPK cascade and TGFβ/SMADs signaling (J Exp Clin Cancer Res.2019, 38(1):178. doi: 10.1186/s13046-019-1176-1).

These issues and above references should be commented in the text by the authors.

4- In recent years, an extensive literature has highlighted some extra anti-hyperglycemic effects of metformin, including anti-aging action (Diabetes Research and Clinical Practice, 2020, 160, 108025. Doi: 10.1016 / j.diabres.2020.108025), and endothelial protection (Biomedicines, 2021, 9 (1), pp. 1-26, 3. doi: 10.3390 / biomedicines9010003), that may share some mechanisms with its anti-cancer role. This interesting issue and the above references should be added in the chapter on metformin.

5- The manuscript needs a linguistic revision of a native English speaker.

Reviewer 2 Report

The article presents an interesting topic and is well structured. However, in view of the aims and scope of Biomedicines.

I would expect in a review a critical appraisal of the epidemiological data, particularly when  conflicted data are presented (e.g at line 157 the affirmation regarding the pancreas cancer is not supported by data fromFig 1, particularly the FRESCO study or the association between thiazolidinediones with bladder cancer). This will improve the value of the article that should not be only a summery of published data.

There are also some minor changes as well:

  • I suppose that Fig 1. represents the hazard ratio for death from different types of cancers in T2D but this is not clear from the title of the figures. Must be reformulated.
  • The title of Table 2 (line 252) is wrongly named Table 3.  

Round 2

Reviewer 1 Report

The authors fully addressed all issues raised by this reviewer

Reviewer 2 Report

No further suggestions